# Sox1a mediates the ability of the parapineal to impart habenular left-right asymmetry

Ingrid Lekk[1,2]*, Véronique Duboc[3,4], Ana Faro[1], Stephanos Nicolaou[1,5], Patrick Blader[3], Stephen W Wilson[1]*

[1]Department of Cell and Developmental Biology, University College London, London, United Kingdom; [2]Center for Brain Research, Medical University of Vienna, Vienna, Austria; [3]Centre de Biologie Intégrative (FR 3743), Centre de Biologie du Développement (UMR5547), Université de Toulouse, CNRS, Toulouse, France; [4]Université Côte d'Azur, CHU, Inserm, CNRS, IRCAN, Nice, France; [5]Division of Cancer Therapeutics, The Institute of Cancer Research, London, United Kingdom

**Abstract** Left-right asymmetries in the zebrafish habenular nuclei are dependent upon the formation of the parapineal, a unilateral group of neurons that arise from the medially positioned pineal complex. In this study, we show that both the left and right habenula are competent to adopt left-type molecular character and efferent connectivity upon the presence of only a few parapineal cells. This ability to impart left-sided character is lost in parapineal cells lacking Sox1a function, despite the normal specification of the parapineal itself. Precisely timed laser ablation experiments demonstrate that the parapineal influences neurogenesis in the left habenula at early developmental stages as well as neurotransmitter phenotype and efferent connectivity during subsequent stages of habenular differentiation. These results reveal a tight coordination between the formation of the unilateral parapineal nucleus and emergence of asymmetric habenulae, ensuring that appropriate lateralised character is propagated within left and right-sided circuitry.
DOI: https://doi.org/10.7554/eLife.47376.001

*For correspondence:
i.lekk.12@ucl.ac.uk (IL);
s.wilson@ucl.ac.uk (SWW)

**Competing interests:** The authors declare that no competing interests exist.

## Introduction

Although once considered to be a mark of cognitive superiority of the human cortex, it is now clear that left-right asymmetries are a consistent feature of all vertebrate brains studied, as well as invertebrate nervous systems (*Alqadah et al., 2018*; *Concha et al., 2012*; *Duboc et al., 2015*; *Frasnelli, 2013*; *Frasnelli et al., 2012*; *Rogers, 2014*). Lateralisation of brain function has many potential advantages, such as sparing energetically expensive brain tissue, decreasing reaction time by avoiding eliciting incompatible responses, providing an advantage in motor learning and facilitating coordinated behaviour in social animals (*Concha et al., 2012*; *Rogers, 2014*; *Rogers and Andrew, 2013*; *Vallortigara and Rogers, 2005*). Not only does evolutionary conservation of brain asymmetries emphasise the importance of hemispheric lateralisation, but it also allows comparative developmental and behavioural studies between species.

Zebrafish (*Danio rerio*) have become an advantageous model in studying brain asymmetries owing to their rapid embryonic development, amenability to genetic manipulation, as well as conveniently small size and transparency for developmental imaging and behavioural analysis. With respect to CNS lateralisation, the focus has long been on the epithalamus which displays overt left-right asymmetries in structure and function not only in zebrafish but in a large number of vertebrates, albeit the extent and laterality of these asymmetries varies greatly between different groups (*Aizawa et al., 2011*; *Bianco and Wilson, 2009*; *Concha and Wilson, 2001*).

The epithalamus is a dorsal subdivision of the diencephalon constituted by bilateral habenular nuclei and a medially positioned pineal complex. The habenula (Hb) is a phylogenetically old brain structure, which functions as a relay station conveying information from the limbic forebrain and sensory systems to the ventral midbrain (*Aizawa et al., 2011*; *Bianco and Wilson, 2009*), whereas the pineal has a conserved role in melatonin release and regulation of circadian rhythms (*Ekström and Meissl, 2003*; *Sapède and Cau, 2013*). The pineal complex also contains an accessory nucleus in some species: a frontal organ in anuran amphibians, a parietal eye in some species of lizards and a parapineal nucleus in jawless and teleost fish (*Concha and Wilson, 2001*).

Epithalamic asymmetries in larval zebrafish manifest at many levels in both the pineal complex and the habenulae. The zebrafish habenulae are divided into dorsal and ventral habenula (dHb, vHb) on both sides, corresponding to mammalian medial and lateral habenula, respectively (*Aizawa et al., 2005*; *Amo et al., 2010*). No overt asymmetries have been described in the zebrafish vHb, whereas the left and right dHb exhibit overt differences in cytoarchitecture and molecular signature (*Concha et al., 2000*; *Concha et al., 2003*; *Gamse et al., 2003*), as well as afferent and efferent connectivity (*Aizawa et al., 2005*; *Bianco et al., 2008*; *Gamse et al., 2005*; *Krishnan et al., 2014*; *Miyasaka et al., 2009*; *Turner et al., 2016*; *Zhang et al., 2017*) and function (*Agetsuma et al., 2010*; *Dreosti et al., 2014*; *Facchin et al., 2015*; *Krishnan et al., 2014*; *Zhang et al., 2017*). The left and right dHb can be further divided into lateral and medial subdomains. The lateral subnucleus (dHbL) is larger on the left side and projects mainly to the dorsal interpeduncular nucleus (IPN), whereas the medial subnucleus (dHbM) is enlarged on the right side and projects exclusively to the ventral IPN (*Aizawa et al., 2005*; *Bianco et al., 2008*; *Gamse et al., 2005*). Therefore, left-right asymmetries in the zebrafish dHb are translated into laterotopic dorsoventral innervation of the midbrain IPN. Comparable organisation appears to be conserved amongst teleost and jawless fish (*Signore et al., 2009*; *Stephenson-Jones et al., 2012*; *Villalón et al., 2012*) but is not obvious in mammals (*Kuan et al., 2007a*). Rather than overt structural asymmetries, mammalian habenular asymmetries manifest at the level of neuronal activity, possibly to allow more flexible lateralisation of habenular circuit function (*Ichijo et al., 2016*). Mammalian habenular asymmetries have also predominantly been observed in the lateral rather than the medial Hb (*Hétu et al., 2016*; *Ichijo et al., 2015*; *Savitz et al., 2011a*; *Savitz et al., 2011b*). It has been hypothesised that the asymmetric connectivity of the dHb in fishes might reflect the division in processing sensory *versus* forebrain contextual input, whereas in mammals such division is lost due to lack of direct sensory input to the epithalamus (*Stephenson-Jones et al., 2012*). The expression of opsins and the presence of photoreceptors in the parapineal suggests that this nucleus might provide such asymmetric sensory input to the epithalamus of teleost fish and lampreys (*Blackshaw and Snyder, 1997*; *Borg et al., 1983*; *Koyanagi et al., 2004*; *Vigh-Teichmann et al., 1983*; *Yáñez et al., 1999*).

In addition to its likely photosensory function, the left-sided parapineal is also essential for the development of most left-right asymmetries in the zebrafish habenulae. Mutants in which the parapineal is not properly specified (*Clanton et al., 2013*; *Regan et al., 2009*; *Snelson et al., 2008*) or in experimental setups where the parapineal is laser-ablated at early developmental stages (*Aizawa et al., 2005*; *Bianco et al., 2008*; *Concha et al., 2003*; *Gamse et al., 2005*; *Gamse et al., 2003*), left dHb characteristics largely fail to develop and the habenulae exhibit right isomerism (a double-right phenotype). One of the mechanisms possibly influenced by the parapineal is the differential timing of neurogenesis in the left and right dHb. As shown by 5-bromo-2-deoxyuridine (BrdU) birth-date analysis, neurogenesis peaks at 32 hpf in the dHbL (more prominent on the left) and at 50 hpf in the dHbM (more prominent on the right) (*Aizawa et al., 2007*). However, the early onset of asymmetric neurogenesis, marked by expression of the neuronal marker *cxcr4b* specifically in the left dHb, can already be detected at 28 hpf and requires left-sided epithalamic Nodal signalling (*Roussigné et al., 2009*). Around that time, left-sided Nodal signalling also determines the direction of parapineal migration – in the case of absent or bilateral epithalamic Nodal signalling, parapineal migration is randomised and consequently habenular asymmetries are reversed in 50% of the embryos (*Aizawa et al., 2005*; *Concha et al., 2000*). Since the asymmetries in Nodal-dependent habenular neurogenesis are very subtle, biasing the migration of the parapineal to the left side might provide a mechanism to further enhance left dHb neurogenesis.

In this study, we address the role of the Sox family transcription factor encoding gene *sox1a* in mediating the ability of the parapineal to influence habenular development. Zebrafish *sox1a* and

*sox1b* have arisen from an ancestral vertebrate *Sox1* gene during teleost genome duplication (*Bowles et al., 2000*) and show largely overlapping expression at early stages from 21 somites to 25 hours post fertilisation (hpf) in the telencephalon, hypothalamus, eye field, early lateral line and otic vesicle primordia, trigeminal placode, lens and spinal cord interneurons (*Gerber et al., 2019*; *Okuda et al., 2006*). However, *sox1a*-specific expression has been detected in the lateral line primordium at 24 hpf (*Gerber et al., 2019*) and in the parapineal from 26 to 28 hpf onwards, but not the pineal anlage from which the parapineal arises (*Clanton et al., 2013*). Hence, Sox1a is a candidate transcription factor for being involved in parapineal specification and/or the role of the parapineal in imparting habenular asymmetry.

Through analysis of the role of *sox1a* in epithalamic development, we find that the parapineal forms and migrates normally in *sox1a*⁻/⁻ mutant zebrafish larvae but the habenulae exhibit right isomerism. Furthermore, transplants of a few wild-type parapineal cells are able to rescue epithalamic asymmetries in *sox1a*⁻/⁻ embryos and induce left-dHb characteristics in both left and right habenula. A time-course of parapineal ablations reveals a previously unsuspected step-wise regulation of habenula development by the parapineal. Our results highlight the essential role of the parapineal and of Sox1a in asymmetric development of adjacent habenula.

## Results

### *sox1a* is expressed in the developing parapineal from the onset of its formation

Whole mount in situ hybridisation analysis showed that *sox1a* is expressed in the parapineal from the onset of its formation between 26 and 28 hpf (*Figure 1A–D"*) (*Clanton et al., 2013*). Fluorescent in situ labelling of *sox1a* mRNA in embryos expressing the Tg(*foxD3:GFP*)$^{zf104}$ and Tg(*flh:eGFP*)$^{U711}$ transgenes in the whole pineal complex (*Concha et al., 2003*) revealed that *sox1a* is first expressed in a few cells located on the left side of the forming parapineal at 28 hpf, and thereafter in all parapineal cells as they undergo collective migration to the left side of the epithalamus (*Figure 2A*). Additionally, *sox1a* is expressed in other areas such as the lens vesicle, anterior forebrain, ventral diencephalon, hindbrain and pharyngeal arches (*Figure 1A–D'*), as has also been described previously (*Gerber et al., 2019*; *Okuda et al., 2006*; *Thisse and Thisse, 2004*).

Parapineal-specific expression of *sox1a* raises two questions: firstly, is *sox1a* function required for parapineal specification and secondly – considering the essential role of the parapineal in elaborating left-sided dHb character (*Aizawa et al., 2005*; *Bianco et al., 2008*; *Concha et al., 2003*; *Gamse et al., 2005*; *Gamse et al., 2003*) – is *sox1a* involved in the regulation of habenular asymmetry?

### The parapineal forms in *sox1a*⁻/⁻ mutants

Using CRISPR/Cas9 genome editing (*Auer et al., 2014a*; *Auer et al., 2014b*; *Gagnon et al., 2014*; *Talbot and Amacher, 2014*), we generated two *sox1a* mutant lines (*Figure 2—figure supplement 1*). The *sox1a*$^{ups8}$ allele (hereafter referred to as *sox1a*⁻/⁻), has an 11 bp deletion in the single exon of the *sox1a* gene, which leads to a premature stop codon at amino acid 62. As a result, the mutant Sox1a protein lacks the HMG DNA binding domain and is predicted to be non-functional. Indeed, no *sox1a* mRNA was detected in the parapineal of mutant embryos (*Figure 2B*) suggesting nonsense-mediated decay of the mutant transcript. The second *sox1a*$^{u5039}$ allele has a 10 bp deletion leading to a premature stop at amino acid 134 leaving the HMG DNA binding domain intact. Both *sox1a*⁻/⁻ mutants show no overt developmental abnormalities and are viable as adults. However, further analyses showed some variable expressivity of the phenotypes described below in the *sox1a*$^{u5039}$ mutant allele and consequently the *sox1a*$^{ups8}$ allele was used for all experiments.

Taking advantage of the Tg(*foxD3:GFP*)$^{zf104}$ and Tg(*flh:eGFP*)$^{U711}$ transgenes to track parapineal cells, we observed that the parapineal migrated with normal timing and trajectory in *sox1a*⁻/⁻ mutants (*Figure 2A–B*). Furthermore, parapineal-specific expression of the transcription factor encoding genes *otx5* (*Gamse et al., 2002*) and *gfi1ab* (*Dufourcq et al., 2004*) was not affected in the *sox1a*⁻/⁻ mutants (*Figure 2C–D*). These results indicate that Sox1a is neither required for parapineal specification nor for migration.

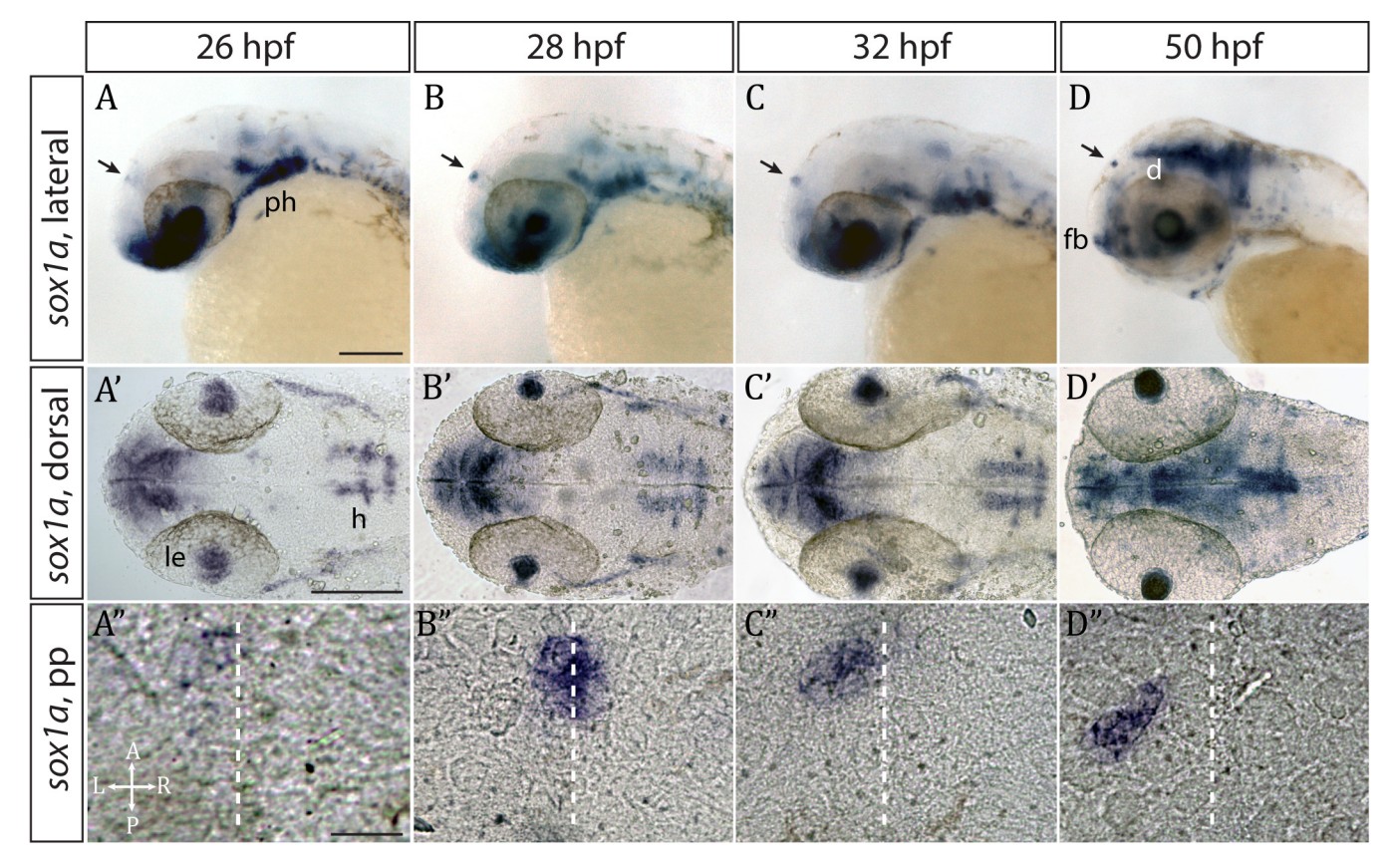

**Figure 1.** *sox1a* is expressed in the parapineal from the onset of its formation. (**A–D'**) Lateral (**A–D**) and dorsal (**A'–D'**) views of zebrafish embryos showing *sox1a* mRNA expression at stages indicated. In addition to the parapineal (indicated by black arrows in **A-D**), *sox1a* is also expressed in the lens vesicle (le), hindbrain (h) and pharyngeal arches (ph). At 50 hpf, *sox1a* is also detected in the ventral diencephalon (**d**) and the anterior forebrain (fb). Scale bars 100 µm. (**A"–D"**) Dorsal views of the epithalamus showing *sox1a* mRNA expression in the migrating parapineal at stages shown above. Dashed line indicates the midline. Scale bars 25 µm.

DOI: https://doi.org/10.7554/eLife.47376.002

Although parapineal neurons form in *sox1a*$^{-/-}$ mutants, efferent projections to the left habenula show reduced outgrowth and branching (**Figure 2E**). At 50 hpf, some parapineal projections could be detected in *sox1a*$^{-/-}$ mutants, albeit with a severely inhibited growth compared to wild-type siblings (**Figure 2—figure supplement 2A**). This further suggests that the initiation of parapineal cell differentiation is not abolished in *sox1a*$^{-/-}$ mutants. Nevertheless, by 4 dpf the parapineal projections were either absent or stunted and lacked branching in all *sox1a*$^{-/-}$ mutant larvae analysed (**Figure 2—figure supplement 2B**). However, this phenotype does not necessarily reflect a cell autonomous deficit in the parapineal neurons as the changes in the left dHb of *sox1a*$^{-/-}$ mutants (see below) are likely to impact its innervation by parapineal axons.

## sox1a$^{-/-}$ mutants and morphants have a double-right dHb similar to parapineal-ablated larvae

Ablation studies have shown that the presence of a parapineal is required for the left dHb to elaborate left-sided patterns of gene expression and connectivity (*Aizawa et al., 2005*; *Bianco et al., 2008*; *Concha et al., 2003*; *Gamse et al., 2005*; *Gamse et al., 2003*). Consequently, we assessed both habenular gene expression and efferent connectivity of habenular neurons in *sox1a*$^{-/-}$ mutants.

Despite normal parapineal specification and migration, *sox1a*$^{-/-}$ mutants have a double-right dHb phenotype (**Figure 3A'–E'**) compared to wild-type siblings (**Figure 3A–E**). Hence, the predominantly left-sided expression of *kctd12.1* (n = 45) (*Gamse et al., 2003*) and *nrp1a* (n = 46) (*Kuan et al.,*

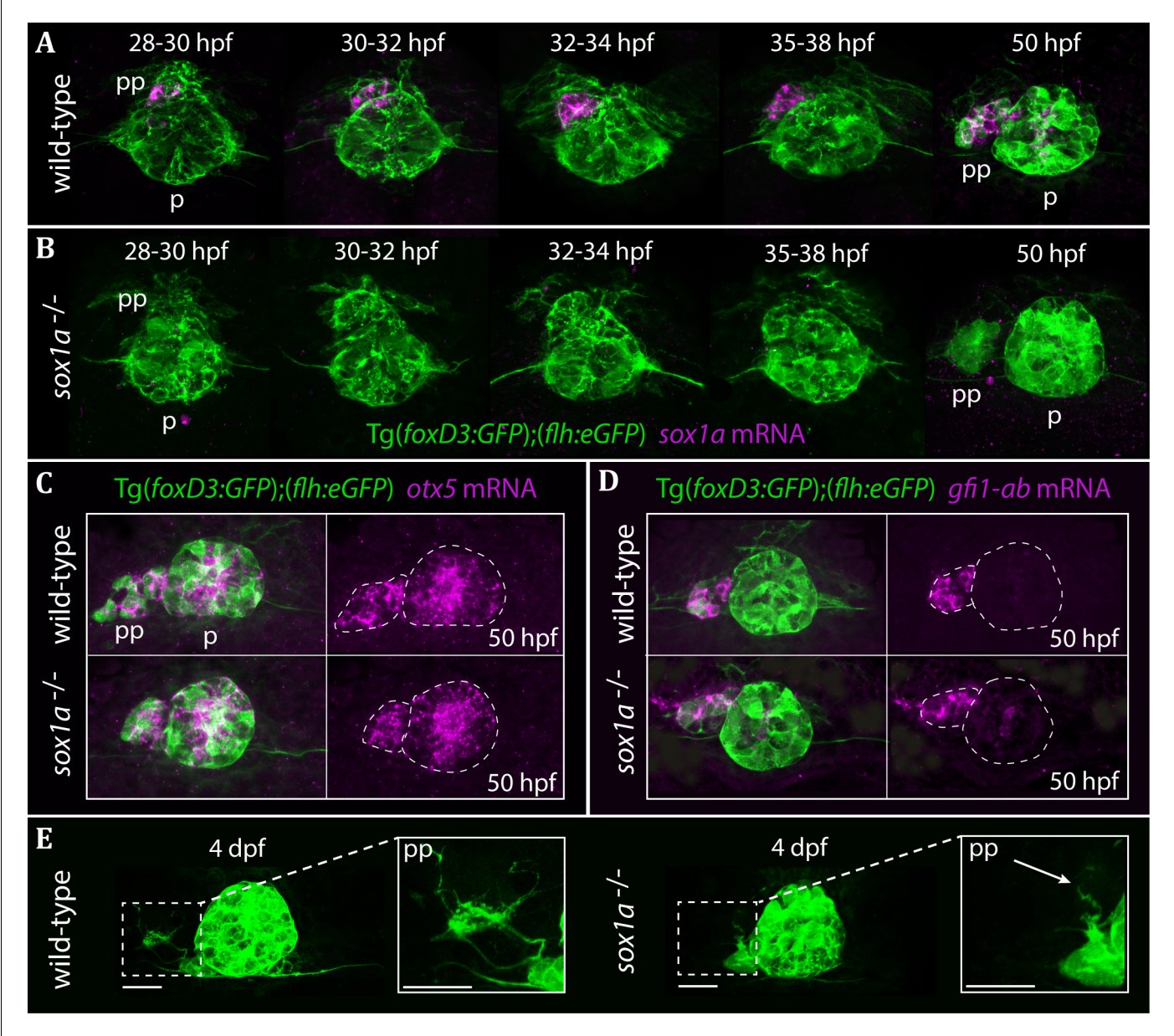

**Figure 2.** The parapineal is specified and migrates normally in *sox1a*⁻/⁻ mutants. All images show dorsal views of the epithalami of wild-type or *sox1a* mutant embryos with expression of Tg(*foxD3:GFP*)$^{zf104}$ and Tg(*flh:eGFP*)$^{U711}$ transgenes (green) in the pineal (p) and the parapineal (pp). mRNA expression of genes indicated is shown in magenta. (**A–B**) Time-course of parapineal migration in Tg(*foxD3:GFP*);(*flh:eGFP*) (**A**) and *sox1a*⁻/⁻ Tg(*foxD3:GFP*);(*flh:eGFP*) (**B**) embryos. Note the absence of *sox1a* mRNA in the parapineal cells of the *sox1a*⁻/⁻ mutants. (**C–D**) *otx5* (pineal and parapineal) and *gfi1ab* (parapineal) mRNA expression in wild-type and *sox1a*⁻/⁻ mutant embryos at 50 hpf. (**E**) Efferent parapineal projections to the left dHb in wild-type and *sox1a* mutant embryos at 4 dpf. Note the stunted projection arising from the *sox1a*⁻/⁻ parapineal (arrow). Scale bars 25 μm.
DOI: https://doi.org/10.7554/eLife.47376.003

The following figure supplements are available for figure 2:

**Figure supplement 1.** Generation of mutant *sox1a* alleles.
DOI: https://doi.org/10.7554/eLife.47376.004

**Figure supplement 2.** Parapineal projections show reduced growth in *sox1a*⁻/⁻ mutants.
DOI: https://doi.org/10.7554/eLife.47376.005

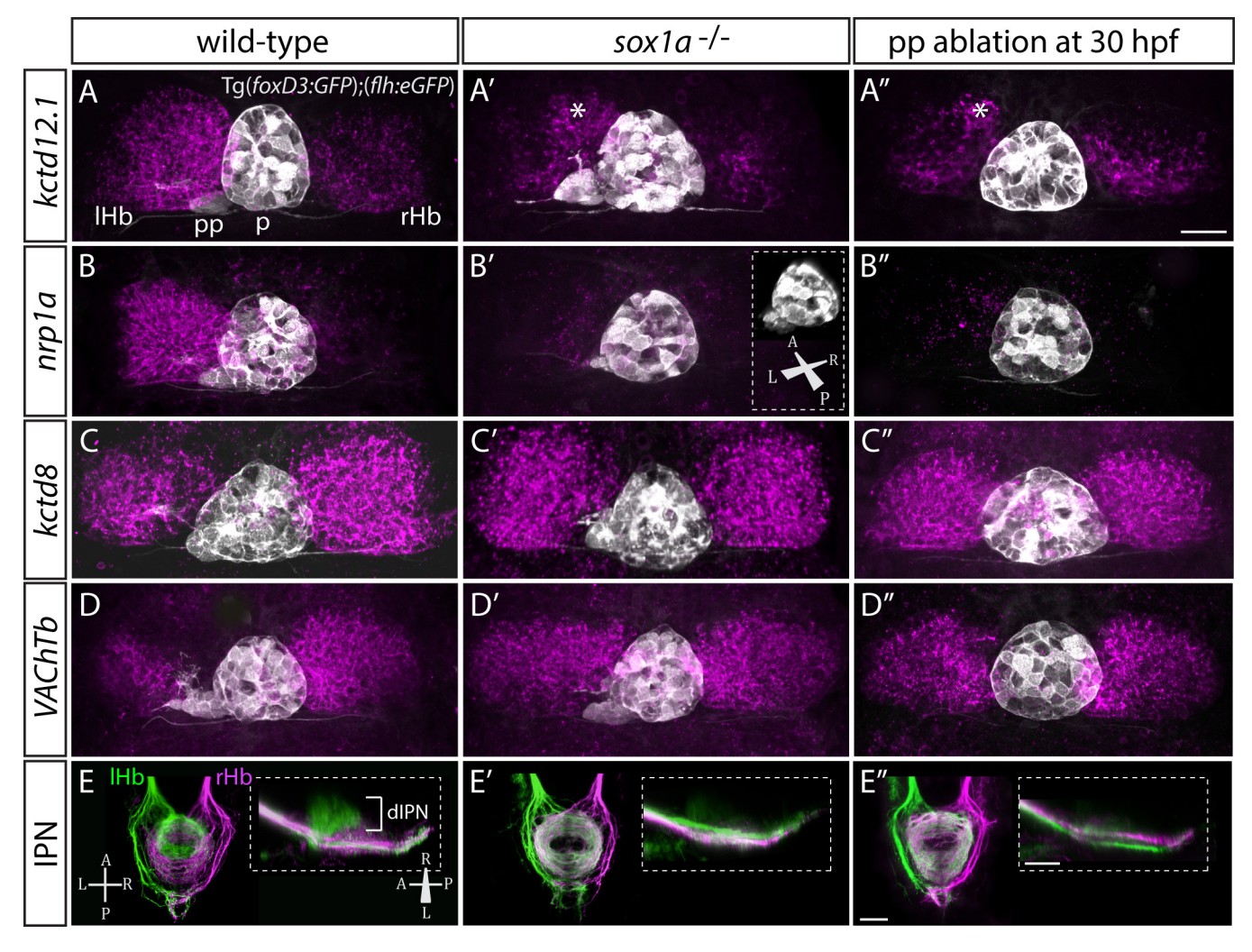

**Figure 3.** *sox1a*[-/-] mutants show a double-right habenular phenotype. (**A–D''**) Dorsal views of the epithalami of wild-type (**A–D**), *sox1a* mutant (**A'–D'**) and parapineal-ablated (**A''–D''**) larvae at 4 dpf showing expression of Tg(*foxD3:GFP*)[zf104] and Tg(*flh:eGFP*)[U711] transgenes (grey) in the pineal (p) and the parapineal (pp). Habenular mRNA expression of genes indicated on the left is shown in magenta (lHb – left habenula, rHb – right habenula, asterisk – residual asymmetry). Scale bar 25 μm. (**E–E''**) Dorsal (left images) and lateral (right images) views of the midbrain interpeduncular nucleus (IPN) labelled by anterograde tracing of axons from left dorsal habenula (lHb, green) and right dorsal habenula (rHb, magenta) at 4 dpf. Note the loss of dorsal IPN (dIPN) innervation by the left habenula in the *sox1a*[-/-] mutant (**E'**) and parapineal-ablated larva (**E''**). Scale bars 25 μm.
DOI: https://doi.org/10.7554/eLife.47376.006

The following figure supplements are available for figure 3:

**Figure supplement 1.** *sox1a* morphants show a double-right habenular phenotype comparable to *sox1a* mutants.
DOI: https://doi.org/10.7554/eLife.47376.007

**Figure supplement 2.** Two *sox1a* mutant alleles do not complement each other.
DOI: https://doi.org/10.7554/eLife.47376.008

*2007a*) was markedly reduced in all *sox1a*[-/-] mutants (*Figure 3A–A' and B–B'*). Conversely, there was increased expression of *kctd8* (n = 46) (*Gamse et al., 2005*) and *VAChTb* (n = 41) (*Hong et al., 2013*), which are normally expressed at higher levels in the right dHb (*Figure 3C–C' and D–D'*). Similar results were obtained for *sox1a* morphants (*Figure 3—figure supplement 1*) and the two *sox1a* mutant alleles (*sox1a*[ups8], *sox1a*[u5039]) failed to complement, with trans-heterozygotes for the two alleles showing the same double-right dHb phenotype as *sox1a*[ups8] homozygotes (*sox1a*[-/-]) (*Figure 3—figure supplement 2*).

The overtly symmetric double-right habenular phenotype in *sox1a*<sup>-/-</sup> mutants and morphants is comparable to the double-right habenulae development upon parapineal ablation at early stages (before parapineal migration at 30 hpf) (*Figure 3A"–D"*) (also previously shown in *Concha et al., 2003*; *Gamse et al., 2003*). This indicates that the forming parapineal in *sox1a*<sup>-/-</sup> mutants is not functional in terms of regulating left dHb development. Note that the residual asymmetry in *kctd12.1* mRNA expression in the dorsomedial domain of left dHb apparent in mutants, morphants and parapineal-ablated larvae alike (asterisk in *Figure 3A'–A"* and *Figure 3—figure supplement 1D'*), is similar to what has previously been described for residual asymmetries in habenular neuropil upon early parapineal ablation (*Bianco et al., 2008*; *Concha et al., 2003*). These asymmetries might be the result of Nodal-dependent neurogenesis in the left dHb, that is potentially independent from parapineal-regulated habenular asymmetries (*Roussigné et al., 2009*).

The symmetric double-right habenular phenotype of *sox1a*<sup>-/-</sup> mutants was also evident in the efferent habenular projections to the IPN, as shown by anterograde axon tracing via lipophilic dye labelling (*Figure 3E–E'*). In *sox1a*<sup>-/-</sup> mutants, dorsal IPN innervation which normally arises from dHbL neurons (more prominent on the left) was almost completely lost and both dHb projected predominantly to the ventral IPN (n = 15), the target of dHbM neurons (*Aizawa et al., 2005*; *Bianco et al., 2008*; *Gamse et al., 2005*). This indicates that in *sox1a*<sup>-/-</sup> mutants, most left dHb neurons have adopted dHbM character similar to the right dHb. Comparable efferent dHb projections predominantly targeting the ventral IPN were observed in early parapineal ablated embryos (*Figure 3E"*), as previously described (*Aizawa et al., 2005*; *Bianco et al., 2008*; *Gamse et al., 2005*), confirming that the parapineal fails to signal to the left habenula in absence of Sox1a function.

In summary, loss of function of the transcription factor Sox1a leads to double-right dHb phenotype similar to parapineal-ablated larvae, despite normal parapineal formation in the mutants.

## Wild-type parapineal cells induce left habenula characteristics in *sox1a*<sup>-/-</sup> mutants

The results described above are consistent with Sox1a in parapineal cells regulating the ability of these cells to impart left-sided character to the left dHb. However, as *sox1a* is expressed elsewhere in and around the nervous system, it is also possible that the habenular phenotype is a consequence of a role for Sox1a outside of the parapineal. To directly test whether Sox1a function is required within the parapineal to elicit habenular phenotypes, we transplanted wild-type parapineal cells or control pineal cells into *sox1a*<sup>-/-</sup> Tg(*foxD3:GFP*); (*flh:eGFP*) embryos at 32 hpf, either to the left or right side of the endogenous pineal complex and assessed dHb character at 4 dpf (*Figure 4*). Subsequent to transplantation, 3–4 transplanted parapineal cells with projections to the adjacent habenula could be detected by live imaging at 50 hpf (*Figure 4A–B and D–E*), whereas transplanted control pineal cells usually re-integrated into the pineal (*Figure 4C and F*).

By four dpf, transplanted wild-type parapineal cells induced left dHb characteristics in the adjacent (left or right) habenula of *sox1a*<sup>-/-</sup> mutants (*Figure 4A'–B' and D'-E'*; n = 11), whereas embryos with pineal-cell transplants still exhibited a double-right dHb phenotype (*Figure 4C' and F'*; n = 5). Hence, while the expression of the left dHb marker *kctd12.1* in *sox1a*<sup>-/-</sup> mutants with pineal-cell transplants (n = 3) was symmetric (double-right) (*Figure 4C'*), *kctd12.1* expression in *sox1a*<sup>-/-</sup> mutants with wildtype parapineal cells on the left side (n = 2) resembled the wild-type condition (higher expression on the left) (*Figure 4A'*) and *kctd12.1* expression in *sox1a*<sup>-/-</sup> mutants with parapineal cells on the right side (n = 2) had a reversed phenotype (higher expression on the right) (*Figure 4B'*). Furthermore, the left or right dHb in *sox1a*<sup>-/-</sup> mutants positioned adjacent to the transplanted wild-type parapineal cells innervated the dorsal IPN (*Figure 4D' and E'*) (n = 4 and n = 3, respectively) comparable to the left dHb of wild type larvae, whereas *sox1a*<sup>-/-</sup> mutants with pineal-cell transplants showed innervation of the ventral IPN from both left and habenula (n = 2) (*Figure 4F'*). Transplanted wild-type parapineal cells sent out extensive axonal projections, as can be distinguished in cells transplanted to the right side where there are no endogenous parapineal cells (white arrows in *Figure 4B–B' and E–E'*).

As expected, transplanted parapineal cells were also able to induce left dHb characteristics in wild-type right habenula, with as few as two parapineal cells being sufficient to change the laterality of the adjacent right dHb (n = 2; *Figure 4—figure supplement 1*). Although it is surprising that so few parapineal cells can have such a large effect, the result is consistent with 'failed' parapineal

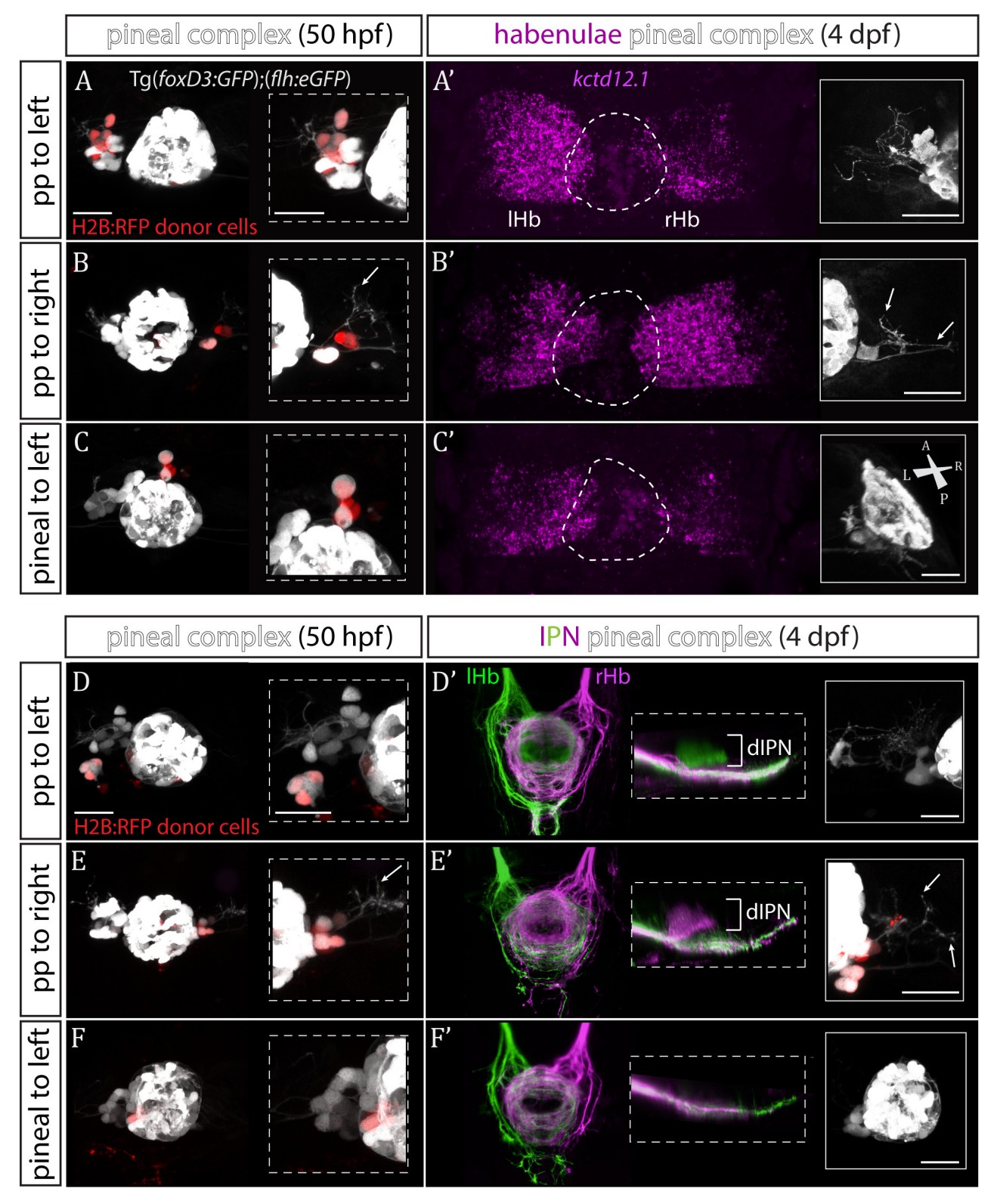

**Figure 4.** Transplanted wild-type parapineal cells rescue habenular asymmetry in *sox1a⁻/⁻* mutants. (**A–F**) Parapineal (pp) cells from an *H2B:RFP* mRNA-injected Tg(*foxD3:GFP*);(*flh:eGFP*) donor embryo transplanted to the left (**A, D**) or right (**B, E**) side of a *sox1a⁻/⁻* Tg(*foxD3:GFP*);(*flh:eGFP*) recipient at 32 hpf send projections to the habenula at 50 hpf (white arrows in B and E), as shown by live-imaging. Transplanted pineal cells (**C, F**) do not send projections to the habenula and locate to the midline (**C**) or reincorporate into the pineal (**F**) by 50 hpf. Scale bars 25 μm. (**A'–F'**) By 4 dpf, the habenula

*Figure 4 continued on next page*

Figure 4 continued

adjacent to the transplanted wild-type parapineal cells acquires a left habenula phenotype in *sox1a*<sup></sup> mutants as shown by *kctd12.1* mRNA expression (**A',B'**) and anterograde labelling of habenula-IPN projections (dorsal and lateral views) (**D',E'**). *sox1a*<sup>-/-</sup> larvae with pineal cell transplant have double-right habenulae (**C',F'**). Solid boxes show the transplanted parapineal cells at 4 dpf, sending out long projections (arrows in **B'** and **E'**) to the adjacent habenula. The whole pineal complex is shown for the pineal cell transplanted larvae (**C',F'**). lHb – left habenula, rHb – right habenula, dIPN – dorsal interpeduncular nucleus. Scale bars 25 μm.
DOI: https://doi.org/10.7554/eLife.47376.009

The following figure supplement is available for figure 4:

**Figure supplement 1.** Parapineal cells impose left habenula character on right dorsal habenula neurons.
DOI: https://doi.org/10.7554/eLife.47376.010

ablation experiments in which only one or two parapineal cells remained. In such embryos, the left dHb still elaborated normal left-sided character (*Figure 5—figure supplement 1B–B'*).

Successful parapineal transplants in both wild-type and *sox1a*<sup>-/-</sup> embryos suggest that the position of the transplanted parapineal cells is not of vital importance in inducing left habenula characteristics (for example, see the anterior position of the transplanted parapineal cells at 4 dpf in *Figure 4—figure supplement 1A'*).

To conclude, *sox1a*<sup>-/-</sup> habenulae are competent to respond to the presence of wild-type parapineal cells and adopt left-type character, confirming that the *sox1a*<sup>-/-</sup> double-right habenular phenotype results from impaired signalling between the parapineal and the left dHb. Furthermore, both left and right habenula are competent to acquire left dHb character in response to parapineal signals, demonstrating that it is the left-sided migration of the parapineal that underlies asymmetric development of the zebrafish epithalamus.

## The parapineal regulates habenular asymmetry at several developmental stages

The temporal progression in the elaboration of habenular asymmetry spans from early asymmetric neurogenesis starting on the left side as early as 28 hpf (*Aizawa et al., 2007*; *Roussigné et al., 2009*) to neuronal differentiation (*deCarvalho et al., 2014*; *Hüsken and Carl, 2013*; *Hüsken et al., 2014*) and establishment of asymmetric connectivity by 4 dpf (*Aizawa et al., 2005*; *Beretta et al., 2017*; *Bianco et al., 2008*; *Carl et al., 2007*; *deCarvalho et al., 2013*; *Gamse et al., 2005*; *Hendricks and Jesuthasan, 2007*; *Kuan et al., 2007b*; *Turner et al., 2016*; *Zhang et al., 2017*). Therefore, the parapineal might regulate the early initiation of neurogenesis in the left dHb, whereby cells are born into an environment that promotes dHbL differentiation, as opposed to later neurogenesis which favours dHbM fate on the right (*Aizawa et al., 2007*). Alternatively or concomitantly, signals from the parapineal might be needed at later stages to impart and/or maintain dHbL specification and/or asymmetric habenular connectivity. To gain insight into which aspects of habenular development are regulated by the parapineal, we carried out laser ablations of the parapineal in Tg (*foxD3:GFP*);(*flh:eGFP*) fish at 30 hpf, 35 hpf, 50 hpf and 3 dpf and studied the effects on the expression pattern of different habenular markers at 4 dpf as well as habenular efferent projections to the IPN (*Figure 5*).

In line with previous studies (*Bianco et al., 2008*; *Concha et al., 2003*; *Gamse et al., 2005*; *Gamse et al., 2003*), early ablations at 30 hpf led to overtly double-right habenula development, as was evident for all studied markers (*kctd12.1*, n = 25; *nrp1a*, n = 6; *VAChTb*, n = 11) (*Figure 5B,G,L*) as well as for efferent projections (n = 3) (*Figure 5Q*). In contrast, parapineal ablations at 35 hpf (n = 16) and 50 hpf (n = 15) did not obviously affect *kctd12.1* expression at 4 dpf (*Figure 5C,D*). Volumetric analysis did reveal a mild reduction in the average volume of the left dHb *kctd12.1* domain upon 35 and 50 hpf ablations compared to controls (data not shown) but the larvae had obvious asymmetric (left-dominant) *kctd12.1* expression patterns compared to double-right 30 hpf ablation phenotype. These results are in line with the fact that the early wave of asymmetric neurogenesis (predominantly in the left dHb) takes place at around 32 hpf (*Aizawa et al., 2007*) and would therefore – if regulated by the parapineal – not be affected by ablations later than 32 hpf.

Surprisingly, however, parapineal ablations at 35 and 50 hpf still led to development of double-right efferent projections to the IPN by 4 dpf (*Figure 5R,S*; n = 14 and n = 12, respectively), potentially as a consequence of the reduction in axon guidance receptor *nrp1* gene expression in the left

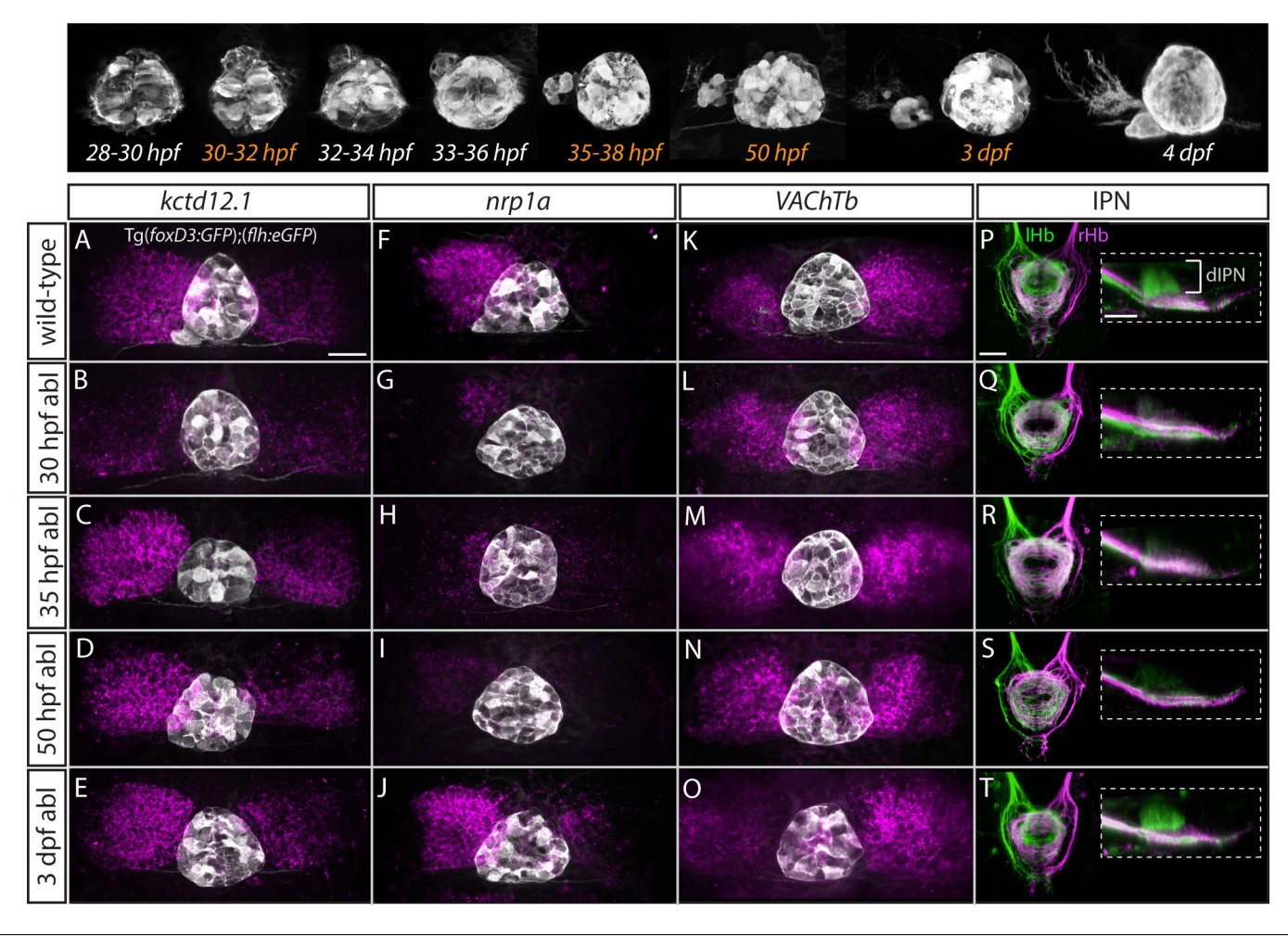

**Figure 5.** Step-wise regulation of habenular asymmetries by the parapineal. The top panel shows the time-course of parapineal development in Tg (*foxD3:GFP*);(*flh:eGFP*) fish. The time-points selected for parapineal ablations are shown in orange. (A–O) Dorsal views of the epithalami of wild-type and parapineal-ablated larvae at 4 dpf showing expression of (*foxD3:GFP*)[zf104] and (*flh:eGFP*)[U711] transgenes (grey) in the pineal complex. Habenular mRNA expression of *kctd12.1*, *nrp1a* and *VAChTb* is shown in magenta. Parapineal ablations were carried out at time-points indicated on the left. Scale bar 25 μm. (P–T) Dorsal (left images) and lateral (right images) views of the interpeduncular nucleus labelled by anterograde tracing of the axons from the left dorsal habenula (lHb, green) and right dorsal habenula (rHb, magenta) at 4 dpf. Parapineal ablations were carried out at time-points indicated on the left. dIPN – dorsal interpeduncular nucleus. Scale bar 25 μm.

DOI: https://doi.org/10.7554/eLife.47376.011

The following figure supplement is available for figure 5:

**Figure supplement 1.** Only ablation of all parapineal cells leads to loss of habenular asymmetries.

DOI: https://doi.org/10.7554/eLife.47376.012

dHb upon 35 and 50 hpf parapineal ablations (*Figure 5H,I*; n = 7 and n = 21, respectively) (*Kuan et al., 2007b*). Furthermore, four dpf *VAChTb* mRNA expression was also affected by 35 hpf (n = 8) and 50 hpf (n = 11) parapineal ablations (*Figure 5M,N*), indicating that asymmetric neurotransmitter domains are not correctly established upon late parapineal ablations. Partial parapineal ablations at 50 hpf or parapineal axotomies did not affect the asymmetry of habenular efferent projections (*Figure 5—figure supplement 1*). Finally, upon parapineal ablations at 3 dpf, all studied habenular asymmetries were of wild-type character at 4 dpf (*Figure 5E,J,O,T*; n = 7, n = 14, n = 15 and n = 6, respectively), consistent with previous data showing that ablation at this stage does not affect lateralised functional properties of habenular neurons (*Dreosti et al., 2014*).

These results indicate that habenular asymmetries are regulated at several developmental stages by the parapineal, firstly at the time of left dHb neurogenesis and thereafter at the level of differentiation (axonal outgrowth and neurotransmitter domains).

## Asymmetric habenular neurogenesis is regulated by the parapineal

The parapineal ablation experiments described above are consistent with the, as yet untested, possibility that the parapineal promotes early, asymmetric neurogenesis in the left dHb. This could contribute to promotion of dHbL character (more prominent on the left), as dHbL neurons tend to be born earlier than dHbM neurons (more prominent on the right) (*Aizawa et al., 2007*). To assess if the parapineal does influence dHb neurogenesis, we carried out BrdU birth-date analysis for dHb neurons in wild-type and parapineal-ablated embryos. Control and 30 hpf parapineal-ablated Tg (*foxD3:GFP*);(*flh:eGFP*) embryos were exposed to a 20 min BrdU pulse at 32 hpf, followed by a chase period until 4 dpf to allow differentiation of the habenular neurons. The number of neurons born around 32 hpf were visualised with BrdU immunofluorescence and habenular asymmetries were assessed by *kctd12.1* in situ hybridisation (*Figure 6A–B'*).

Birthdating analysis demonstrated that the parapineal does influence neurogenesis in the left dHb. In control wild-type embryos (n = 22), significantly more cells were born in the left dHb compared to the right at 32 hpf as expected (p<6×10$^{-5}$, Wilcoxon signed rank test) (*Figure 6E*). This asymmetry between the left and right dHb was markedly reduced in parapineal-ablated embryos (n = 18, p=0.002, Wilcoxon signed rank test), due to decreased neurogenesis in the left dHb

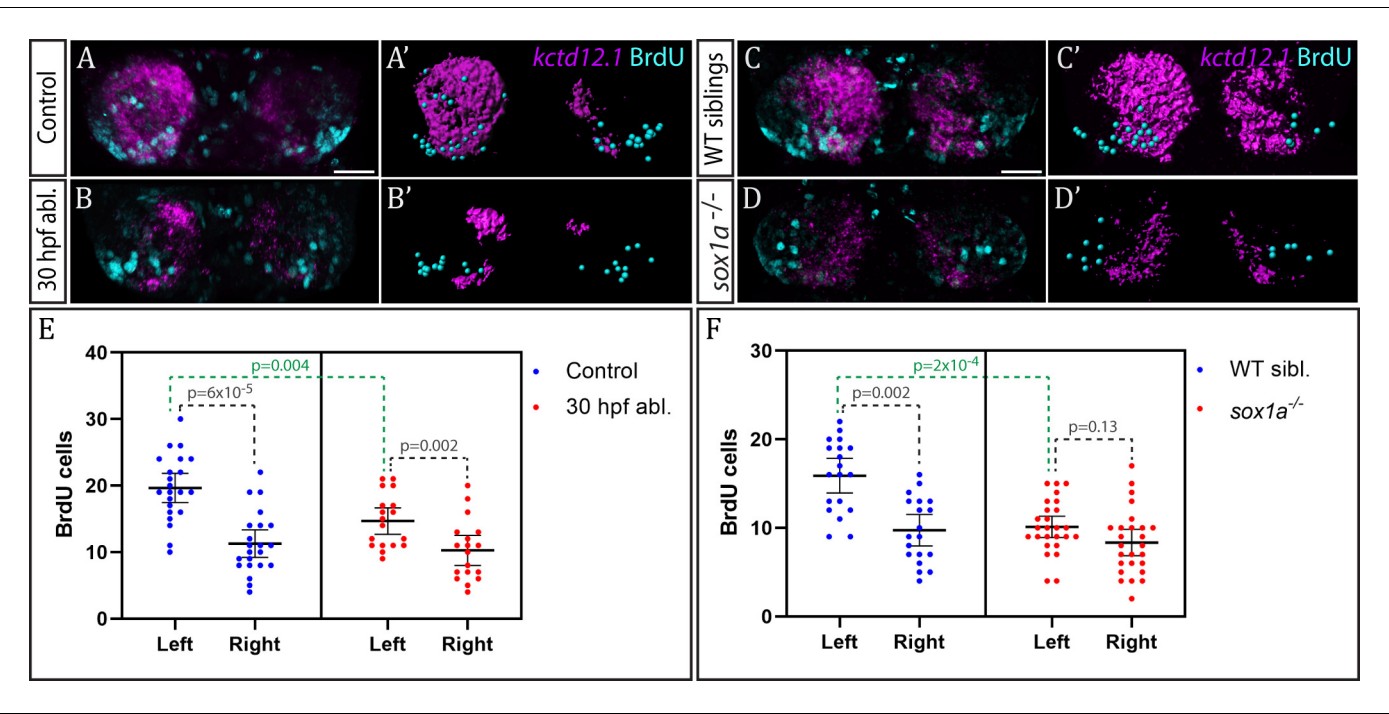

**Figure 6.** Neurogenesis is reduced in the left habenula upon early parapineal ablation and in *sox1a*$^{-/-}$ mutants. (A–D') Confocal images (A–D) and reconstructions (A'–D') of *kctd12.1* mRNA in situ hybridisation (magenta) and BrdU immunohistochemistry (cyan) in the dHb of control (A, A') and parapineal-ablated larvae (B, B'), as well as *sox1a*$^{-/-}$ mutants (D, D') and wild-type siblings (C, C') at 4 dpf. Ablations were carried out at 30 hpf and BrdU pulse given at 32 hpf. Scale bar 25 µm. (E–F) Plots showing the number of BrdU-positive cells born at 32 hpf in the left and right dHb. (E) Control (n = 22) and parapineal-ablated (n = 18) embryos. (F) *sox1a*$^{-/-}$ mutants (n = 26) and wild-type siblings (n = 19). Cells were counted as shown in A'-D', mean and 95% CI are shown. The reduction in left dHb BrdU-positive cells upon 30 hpf ablation (p=0.004, Wilcoxon-Mann-Whitney test) and in *sox1a*$^{-/-}$ mutants (p=2×10$^{-4}$) is indicated in green. Counts for left and right habenula of each individual embryo are given in *Figure 6—source data 1*.
DOI: https://doi.org/10.7554/eLife.47376.013

The following source data is available for figure 6:

**Source data 1.** BrdU-positive cell counts in wild-type, parapineal-ablated and *sox1a*$^{-/-}$ embryos.
DOI: https://doi.org/10.7554/eLife.47376.014

compared to controls (p=0.004, Wilcoxon-Mann-Whitney test), while the right habenula was unaffected (p=0.478, Wilcoxon-Mann-Whitney test) (*Figure 6E*). Concomitantly, *kctd12.1* expression revealed the expected double-right dHb phenotype in the parapineal-ablated embryos (*Figure 6B*).

We also performed analogous birthdate analysis in *sox1a*[-/-] mutants at 32 hpf (*Figure 6C–D'*) and observed results similar to parapineal-ablated embryos. Compared to wild-type siblings, neurogenesis in the left dHb was reduced in *sox1a*[-/-] mutants (p=$2\times10^{-4}$, Wilcoxon-Mann-Whitney test), whereas the right habenula was unaffected (p=0.276, Wilcoxon-Mann-Whitney test). Therefore, there was no significant difference in the 32 hpf BrdU labelling between the left and right dHb of *sox1a*[-/-] mutants (p=0.13, Wilcoxon signed rank test), with both habenula being comparable to the right dHb of wild-type siblings (*Figure 6F*).

These results demonstrate that the parapineal is required for the early wave of neurogenesis that is more prominent in the left dHb. The residual asymmetry in the number of BrdU-positive cells between the left and right dHb in parapineal-ablated embryos and sox1a[-/-] mutants most likely results from a Nodal-dependent (and parapineal-independent) influence upon neurogenesis (*Roussigné et al., 2009*).

## Discussion

Using two alternative approaches – cell ablations/transplants and genetic manipulation – we have shown that epithalamic asymmetries in zebrafish are determined by the unilateral parapineal nucleus, refining and extending previous studies that have drawn similar conclusions. Parapineal cells are able to induce left habenula characteristics in both left and right habenula and in fish lacking function of the transcription factor Sox1a, this inductive ability of the parapineal is lost.

### The parapineal regulates step-wise development of habenular asymmetries

By means of precisely timed laser-ablations, this study has revealed that the parapineal regulates several steps of habenula development. Early parapineal ablations (30 hpf) resulted in double-right dHb, whereas late parapineal ablations (at 35 hpf and 50 hpf) led to loss of some differentiated dHb characteristics (lateralised *nrp1a* and *VAChT* expression, laterotopic efferent connectivity) but not others (*kctd12.1* expression).

BrdU labelling demonstrated that at early stages of development, the parapineal promotes neurogenesis in the left dHb. In comparison with more basal vertebrates, this adds a second mechanism contributing to asymmetric neurogenesis in the habenulae in addition to left-sided Nodal signalling (*Roussigné et al., 2009*). Indeed, habenular neurogenic asymmetries in catshark are regulated by left-sided Nodal signalling (*Lagadec et al., 2018*) and *ktcd* genes in lamprey and catshark habenula are asymmetrically expressed independently from the parapineal, which in catshark is not even present (*Lagadec et al., 2015*). However, in zebrafish, Nodal plays only a minor role in the early onset of asymmetric habenular neurogenesis prior to the formation of the parapineal (*Roussigné et al., 2009*), whereas the major wave of left-sided habenular neurogenesis takes place at 32 hpf (*Aizawa et al., 2007*) and requires the migrating parapineal (this study). The diminished role of Nodal in the development of habenular asymmetry in zebrafish is also apparent from parapineal transplant experiments in which parapineal cells induced left habenula characteristics on the right side (this study), where the Nodal pathway is not activated (*Bisgrove et al., 2000*; *Concha et al., 2000*).

By carrying out parapineal ablations at later stages at (35 hpf and 50 hpf), we have shown that in zebrafish, the parapineal is also required for the development of left dHb characteristics (connectivity, neurotransmitter phenotype) independent of neurogenesis (32 hpf). This is in accordance with previous ablation studies suggesting an additional role of the parapineal in dHb development after left dHb neurogenesis has taken place. Firstly, parapineal ablations at 2–3 days lead to loss of asymmetries in the afferent innervation of the habenulae from the olfactory bulb, with left dHb receiving input in addition to the right (*deCarvalho et al., 2013*). Furthermore, adult fish that have undergone parapineal ablations at 3 dpf show reduced exploratory behaviour (increased anxiety) compared to wild-type siblings, a phenotype likely to result from disruptions in the dHb (*Agetsuma et al., 2010*; *Facchin et al., 2015*). It is possible that the parapineal regulates at least some aspects of left dHb differentiation through modulation of Wnt signalling, as disrupting this pathway alters habenular

lateralisation without overt differences in parapineal development (*Carl et al., 2007*; *Hüsken et al., 2014*).

By transplants and partially effective ablations, we have shown that parapineal cells are remarkably potent at inducing left dHb character in both, left and right habenula with only a few cells being sufficient. It is not clear whether parapineal cells exhibit any heterogeneity regarding their ability to induce left dHb character nor whether direct cell-cell contact is required between the parapineal and the left dHb. A recent study demonstrated focal activation of FGF-signalling in leading cells of the migrating parapineal, indicating that cells possess positional identity within the parapineal (*Roussigné et al., 2018*). This is supported by our observation that the onset of *sox1a* expression in the migrating parapineal is asymmetric, starting at the left (leading) side. In accordance with parapineal migration being a collective behaviour (*Roussigné et al., 2018*), our parapineal transplant experiments suggest that single parapineal cells do not migrate effectively to their final destination but rather stay close to the site of transplantation. However, regardless of the transplant position, left dHb character is nevertheless induced. Taking these observations into account, it is unlikely that left dHb character is induced by discrete parapineal cells contacting a subset of habenular cells during their migration but perhaps rather by a paracrine secreted signal. Indeed, when the parapineal is increased in size, it can impart left-sided character to right-sided habenular neurons despite it being unlikely that the parapineal ever contact these habenular cells (*Garric et al., 2014*).

## The role of the parapineal in the evolution of habenular asymmetries

In light of the above observations, it is tempting to hypothesise that during teleost evolution, the left-sided parapineal has become a dominant signalling centre in the regulation of habenular asymmetries, whereas left-sided Nodal pathway activation is primarily required for determination of laterality (left-sided migration of the parapineal [*Concha et al., 2000*; *Concha et al., 2003*; *Gamse et al., 2003*] and parapineal size [*Garric et al., 2014*]). In this scenario, the parapineal has subsumed an ancestral role of Nodal in the regulation of habenular neurogenesis, possibly due to restrictions in developmental timing and duration of Nodal cascade activity (*Signore and Concha, 2017*; *Signore et al., 2016*).

The parapineal might also have become essential in adding complexity to asymmetries in the habenulae in fishes, namely by regulating the establishment of asymmetric neurotransmitter domains and/or habenular connectivity. To date, laterotopic innervation of the IPN from the medial (teleost dorsal) habenula has only been clearly demonstrated in species with an apparent parapineal nucleus (jawless and teleost fish) (*Aizawa et al., 2005*; *Bianco et al., 2008*; *Gamse et al., 2005*; *Signore et al., 2009*; *Stephenson-Jones et al., 2012*; *Villalón et al., 2012*), even though various degrees of asymmetry in habenular size and subnuclear organisation are present in species representative of most vertebrate classes (*Concha and Wilson, 2001*; *Roussigne et al., 2012*). Hence, the yet to be discovered function of the unilateral connections between the parapineal and the left dHb might have co-evolved with mechanisms regulating the development of habenular efferent connectivity, whereby the parapineal ensures the downstream propagation of lateralised habenular circuity. Currently, the role of the parapineal in the regulation of habenular circuitry development in various fishes other than teleosts remains elusive but such knowledge would greatly enhance our understanding of molecular and cellular mechanisms underlying evolutionary changes in vertebrate brain lateralisation.

## *sox1a*[-/-] mutants have symmetric habenulae with largely double-right character

Zebrafish have long been an excellent model to study genetic regulation of brain asymmetry from development to function (*Concha et al., 2012*; *Concha et al., 2009*; *Duboue and Halpern, 2017*; *Roussigne et al., 2012*). Taking advantage of this model, we have shown that the double-right habenula phenotype of *sox1a*[-/-] mutants is identical to that in parapineal-ablated larvae, revealing a genetic factor behind the development of epithalamic asymmetries in zebrafish. Furthermore, the normal formation and migration of the parapineal in *sox1a*[-/-] mutants despite loss of Sox1a in the parapineal indicates that Sox1a specifically functions in the regulation of signalling between the parapineal and left dHb rather than in parapineal specification.

Despite broad expression of *sox1a* in the embryonic zebrafish brain, *sox1a*[-/-] mutants do not seem to have severe developmental defects other than loss of dHb asymmetry. However, the other teleost *sox1* paralogue – *sox1b* – has a nearly identical expression pattern with *sox1a* at early stages with the exception of the parapineal (*Gerber et al., 2019*; *Okuda et al., 2006*), and 80% sequence similarity with *sox1a* in the ORF, suggesting that these two *sox1* genes are likely to have redundant functions in the developing CNS. Likewise, redundant functions of B1 sox genes have been described for the *Sox1* knock-out mouse, in which formation of the lens (where only *Sox1* is expressed) is severely disrupted, whereas the CNS shows only mild developmental abnormalities (due to overlapping expression of *Sox1* with *Sox2* and *Sox3*) (*Ekonomou et al., 2005*; *Malas et al., 2003*; *Nishiguchi et al., 1998*). The functional redundancy between B1 sox genes has also been suggested in early embryogenesis of zebrafish by combinations of *sox2/3/19a/19b* knock-downs (*Okuda et al., 2010*).

In *sox1a*[-/-] mutants, the parapineal forms normally, although parapineal cells do not send fully developed projections to the left habenula. Rather than a cell-autonomous phenotype, this is most likely a secondary effect due to left habenula character not being specified. Indeed, previous studies have shown that lateralised afferent innervation of the dHb depends on the lateralised character of the left and right dHb (*deCarvalho et al., 2013*; *Dreosti et al., 2014*). It is also unlikely that any of the double-right dHb characteristics described here for *sox1a*[-/-] mutant larvae are caused by the abnormal extension of parapineal axons as parapineal axotomies have no apparent effect on asymmetric habenular efferent connectivity.

The parapineal-specific expression of *sox1a* and the overlapping expression of different B1 *sox* genes in the rest of the zebrafish CNS renders the *sox1a*[-/-] mutant a valuable model for studying genetic regulation of brain asymmetry development in a context without overt defects in other aspects of brain development.

## Conclusions

Here, we have shown that the parapineal is essential for the development of habenular asymmetries in the larval zebrafish at several stages. *sox1a* mutant fish exhibit an almost complete loss of left habenula characteristics despite the formation of a parapineal nucleus providing an excellent genetic tool to study the signalling events responsible for establishing habenular asymmetries. In addition, precise, time-controlled parapineal ablation and transplant experiments demonstrate the step-wise manner of habenula asymmetry regulation by the parapineal and the remarkable potency of parapineal cells to induce left habenula characteristics in both left and right habenulae.

## Materials and methods

**Key resources table**

| Reagent type (species) or resource | Designation | Source or reference | Identifiers | Additional information |
| --- | --- | --- | --- | --- |
| Gene (*Danio rerio*) | *sox1a* | NA | Ensembl ENSDAR G00000069866.5 | Line maintained at S Wilson lab |
| Strain, strain background (*Danio rerio*, AB/TL) | *sox1a*[u5039] | this paper | | Line maintained at S Wilson lab |
| Strain, strain background (*Danio rerio*, AB/TL) | *sox1a*[ups8] | this paper | | Line maintained at S Wilson lab |
| Strain, strain background (*Danio rerio*, AB/TL) | Tg(foxD3:GFP)[zf104] | PMID: 12062041 | | Line maintained at S Wilson lab |
| Strain, strain background (*Danio rerio*, AB/TL) | Tg(flh:eGFP)[U711] | PMID: 12895418 | | Line maintained at S Wilson lab |

*Continued on next page*

*Continued*

| Reagent type (species) or resource | Designation | Source or reference | Identifiers | Additional information |
|---|---|---|---|---|
| Strain, strain background (*Danio rerio*, AB/TL) | Et(*gata2a:eGFP*)$^{pku588}$ | PMID: 18164283 | | |
| Antibody | Rabbit polyclonal anti-GFP | Torrey Pines Biolabs | Cat# TP401 | (1:1000) |
| Antibody | Mouse monoclonal anti-ascetylated tubulin | Sigma | Cat# T7451 | (1:1000) |
| Antibody | Mouse monoclonal anti-BrdU | Roche | Cat# 11170376001 | (1:400) |
| Antibody | Goat polyclonal Alexa 488, 568 and 647-conjugated secondary | Molecular Probes | Cat# A32731, A21144, A21126 | (1:250) |
| Other | DAPI stain | Molecular Probes | | (1:1000) |
| Recombinant DNA reagent (plasmid) | Cas9 plasmid | PMID: 23918387 | NA | Provided by W Chen lab |
| Commercial assay or kit | mMESSAGE mMACHINE Kit | Ambion | Cat# AM1344 | |
| Commercial assay or kit | HiScribe T7 High-Yield RNA Synthesis Kit | New England BioLabs | Cat# E2040S | |
| Commercial assay or kit | Precision Melt Supermix | Bio-Rad | Cat# 172–5112 | |
| commercial assay or kit | KASP chemistry for *sox1a* genotyping | LGC Genomics | NA | Assay designed by manufacturer |
| Software, algorithm | CHOPCHOP | PMID:24861617 | https://chopchop.cbu.uib.no/ | |

## Fish lines and maintenance

Zebrafish (*Danio rerio*) were maintained in the University College London Fish Facility at 28°C and standard light conditions (14 hr light/10 hr dark). Embryos were obtained from natural spawning, raised at 28.5°C and staged as hours or days post fertilisation (hpf, dpf) according to *Kimmel et al. (1995)*. 0.003% 1-phenyl-2-thiourea (PTU) was added to the water at 24–26 hpf to prevent pigmentation. For live-imaging, 0.04 mg/ml (0.02%) Tricaine (ethyl 3-aminobenzoate methanesulfonate) (Sigma) was added to the water for anaesthesia. Previously established fish lines used in this study were Tg(*foxD3:GFP*);(*flh:eGFP*) (*Concha et al., 2003*) from incross of Tg(*foxD3:GFP*)$^{zf104}$ and Tg(*flh:eGFP*)$^{U711}$ (*Concha et al., 2003*; *Gilmour et al., 2002*) and Et(*gata2a:eGFP*)$^{pku588}$ (*Wen et al., 2008*).

## Fixation and dissection

Embryos and larvae were fixed in 4% (w/v) paraformaldehyde (PFA) in phosphate buffered saline (PBS) by over-night immersion at 4°C. For BrdU immunohistochemistry and lipophilic dye labelling, the brain of 4 dpf larvae was dissected out by manual dissection with larvae pinned in sylgard (*Turner et al., 2014*).

## Generation of *sox1a⁻ᐟ⁻* mutants by CRISPR/cas9

*sox1a* mutant lines were generated by CRISPR/Cas9 targeted genome editing relying on non-homologous end joining repair mechanism, as described in detailed protocols provided by *Auer et al. (2014b)*, *Gagnon et al. (2014)*, and *Talbot and Amacher (2014)*. Optimal target sites were selected using the CHOPCHOP web tool (*Montague et al., 2014*). *cas9* mRNA was transcribed from a

plasmid provided by *Jao et al. (2013)* using Ambion mMESSAGE mMACHINE Kit. Guide RNAs were generated by PCR and transcribed using HiScribe T7 High-Yield RNA Synthesis Kit (NEB). 110–140 pg of guide RNA and 170 pg of *cas9* mRNA per embryo was injected at one-cell stage into the cell. Mutants were screened by high-resolution melt analysis (HRMA) (*Dahlem et al., 2012*) using Biorad Precision Melt Supermix and confirmed by Sanger sequencing. The mutated sequences are shown in *Figure 2—figure supplement 1*. Mutants were genotyped for all further experiments by allelic discrimination via KASP chemistry using PCR primers designed by the manufacturer (LGC Genomics) and the CFX Connect Real-Time PCR Detection system (BIO-RAD) for detection and analysis.

## Whole-mount in situ hybridisation (ISH, FISH)

Digoxygenin (Roche) labelled RNA probes were made using standard protocols and spanned a minimum of 800 bp. To enhance permeabilisation, fixed embryos or larvae were dehydrated in methanol for a minimum of one hour at −20°C, rehydrated in PBST (PBS with 0.5% Tween-20, Sigma) and treated with 0.02 mg/ml proteinase K (PK, Sigma) for 10–40 min depending on the age of the fish. Probe hybridisation was carried out at 70°C in standard hybridisation solution containing 50% formamide over-night, with 2 ng/µl of RNA probe. Embryos were washed at 70°C through a graded series of hybridisation solution and 2x saline sodium citrate (SSC), followed by further washes with 0.2x SSC and PBST at room temperature. Blocking was carried out in maleic acid buffer (150 mM maleic acid, 100 mM NaCl, 2% sheep serum, 2 mg/ml BSA) for 2–3 hr. Probes were detected by over-night incubation with anti-Digoxigenin-AP Fab fragments (1:5000) (Roche) and stained with standard Nitro Blue Tetrazolium (NBT) and 5-Bromo-4-chloro-3-indolyl phosphate (BCIP) (Roche) ISH protocol. Fluorescent in situ hybridisation (FISH) was carried out using either Fast Red tablets according to manufacturer's instructions (Roche, discontinued from manufacturing) or Fast Blue BB Salt (Sigma) and NAMP (Sigma) staining as previously described (*Lauter et al., 2011*).

## Whole-mount immunohistochemistry

Fixed larvae were stained and imaged as whole-mounts following standard procedures (*Shanmugalingam et al., 2000*; *Turner et al., 2014*). In short, samples were dehydrated in methanol for a minimum of one hour at −20°C, rehydrated in PBST and treated with 0.02 mg/ml proteinase K (PK, Sigma) for 10–40 min depending on the age of the fish. 10% Heat-inactivated Normal Goat Serum (NGS) (Sigma) was used for block and for over-night primary antibody incubation at 4°C with the following antibodies: rabbit anti-GFP (dilution 1:1000, Torrey Pines Biolabs, Cat# TP401), mouse anti-acetylated tubulin (dilution 1:250, IgG2b, α-tubulin, Sigma Cat# T7451) and mouse monoclonal anti-BrdU antibody (1:450, Roche, Cat# 11170376001). Secondary antibody incubation was carried out over night at 4°C using Alexa Fluor 488-conjugated, 568-conjugated and 647-conjugated secondary antibodies (1:200, Molecular Probes, Cat# A32731, A21144, A21126). For immunohistochemistry after in situ hybridisation, probe hybridisation was carried out at a lower temperature (65–68°C) to ensure high-quality immunolabelling. After Fast Red or Fast Blue in situ hybridisation, samples were washed 6 × 20 min in PBST followed by primary antibody incubation in PBST without NGS and immunohistochemistry as usual.

## Neural tract tracing

Tracing of habenula efferent projections was carried out by labelling with membrane-bound lipophilic dyes DiI (DiIC18(3), Molecular Probes, Cat# D3911) and DiD (DiIC18(5), Molecular Probes, Cat# D7757) in 4 dpf embryos. To that end, immobilised embryos (pinned down from the body with needles) were dissected to expose the brain. For a dorsal view, embryos were then placed between two needles and under a stereomicroscope, crystals of DiI (left dHb) and DiD (right dHb) were manually applied to dorsal habenulae with electrolytically sharpened tungsten needles. Brains were incubated in PBS overnight at 4°C, mounted in 1.5% low melting point agarose (Sigma) in PBS and imaged by confocal laser scanning microscopy. The success rate of bilateral labelling was approximately 60%.

## Parapineal transplants

Tg(*foxD3:GFP*);(*flh:eGFP*) donor embryos were pressure-injected with in vitro transcribed *H2B-RFP* (*histone 2B-RFP*) mRNA at one-cell stage. At 30–32 hpf, parapineal cells from donor embryos were needle aspirated (outer diameter of the needle 1.5 µm) and transplanted into Tg(*foxD3:GFP*);(*flh: eGFP*) or Tg(*sox1a⁻/⁻*);(*foxD3:GFP*);(*flh:eGFP*) recipient embryos of the same stage using a CellTram Vario oil-based manual piston pump (Eppendorf). For this, donor and recipient embryos were manually dechorionated and mounted on a glass slide in 1.5% low melting point agarose (Sigma) in fish water with 0.04 mg/ml Tricaine for anaesthesia. At 50 hpf, the transplants were live-imaged with a two-photon microscope – embryos with transplanted cells adjacent to the pineal were raised to 4 dpf and fixed in 4% PFA (w/v) in PBS at 4°C overnight for further analysis. The n-number of parapineal transplant experiments is limited by the low success rate – at 30–32 hpf the neuroepithelium is rather thin and transplanted cells often fall into the ventricle. Out of the successful transplants, approximately 10% can be detected the following day and further analysed. The survival rate of the embryos that go through the transplantation procedure is 100%.

## Parapineal laser-ablations and axotomies

Two-photon laser-ablations of the parapineal cells and for parapineal axotomies were carried out in Tg(*foxD3:GFP*);(*flh:eGFP*) embryos with double-transgenic GFP expression in the pineal complex (*Concha et al., 2003*), using the Leica 25x/0.95 NA PL FLUOROTAR water-dipping objective on a Leica TCS SP8 Confocal microscope coupled with a multiphoton system (Chameleon Compact OPO-Vis, Coherent) and an environmental chamber at 28.5°C. Embryos were manually dechorionated and immobilised by mounting on a glass slide in a drop of 1.5% low melting point agarose (Sigma) in fish water with 0.04 mg/ml Tricaine for anaesthesia. Ablations were carried out at 2–3 separate z-planes, using 30–60% of the maximum output laser power (80 mW) at the wavelength of 910 nm. Each scan took on average 5–10 s per z-plane. For 30 hpf parapineal ablations, 1/3 of the pineal complex anlage was removed from its anterior end, the position of parapineal precursors (*Concha et al., 2003*). Embryos were removed from agarose directly after the ablations. Ablation success was confirmed by live confocal imaging the next day. The success rate for parapineal two-photon ablations (all cells ablated) is approximately 80% at 35 and 50 hpf but lower (60%) for 30 hpf ablations (due to regeneration of the parapineal) and 3 dpf ablations (possibly due to the compact structure of the parapineal and the blood vessels covering it). Each experiment was carried out in two to three separate replicates with the exception of previously published results (indicated where appropriate), which were confirmed once. The n-numbers for all ablation experiments are given in the Results section. Ablated embryos were analysed with a comparable number of control embryos (embryos mounted in agarose but not ablated).

Axotomies were performed at 30–40% laser power by 2–3 pulses at 910 nm using the Bleach Point function on Leica Application Suite X (LAS X) software. The axon bundle was severed approximately 10 µm from the cell body at three time-points (due to regeneration) – at 50, 60 and 72 hpf. The embryos were removed from agarose in between these time points to ensure normal development. Axotomy success was confirmed by live confocal imaging of the pineal complex at 4 dpf before fixation and dissection for lipophilic dye labelling.

## BrdU birth-date analysis

5-Bromo-2-deoxyuridine (BrdU) labelling was carried out in in Tg(*foxD3:GFP*);(*flh:eGFP*) wild-type and parapineal-ablated embryos, as well as in Tg(*sox1a⁻/⁻*);(*foxD3:GFP*);(*flh:eGFP*) mutants and wild-type siblings. Parapineal ablations were performed at 30 hpf as described above, after which parapineal-ablated embryos and non-ablated controls were immediately recovered from agarose. At 32 hpf, embryos were subjected to BrdU labelling. Embryos were incubated in Claw 1xE3 embryo medium (5 mM NaCl, 0.17 mM KCl, 0.33 mM CaCl$_2$, 0.33 mM MgSO$_4$) with 15% DMSO (Sigma, Cat# 276855) for 5 min on ice, followed by 20 min 10 mM BrdU (Sigma, Cat# B5002) incubation in Claw 1xE3 embryo medium with 15% DMSO on ice. At 50 hpf, parapineal-ablated embryos and non-ablated controls were mounted in agarose and ablation success was confirmed by live confocal imaging. Fail-ablated embryos (with one or more parapineal cell left unablated and/or with a damaged pineal) were excluded from further analysis. At 4 dpf, larvae were fixed and dissected to expose the brain. Brain dissection also leads to loss of the superficially positioned pineal complex in most cases. Fast Red (Roche,

discontinued from manufacturing) fluorescent in situ hybridisation for *kctd12.1* followed by BrdU immunohistochemistry was carried out as described, with an added step of 45 min 2N HCl treatment to expose the BrdU epitope prior to antibody staining. BrdU-positive cells from 3D reconstructions were counted using semi-automated detection in Imaris 7.7.1 (Bitplane) software. The experiment was repeated three times and the results were analysed using GraphPad Prism 8.0.2 software. The n-numbers were limited by high technical difficulty of combining parapineal ablations and BrdU staining in a sort window of time (30–32 hpf), allowing recovery between the two experiments. The data did not show clear normal distribution and therefore non-parametric tests were used for statistical analysis. Wilcoxon-Mann-Whitney test was carried out for unpaired comparisons of BrdU cell counts between control and parapineal-ablated embryos. Wilxocon signed rank test was used for paired analysis of BrdU cell counts in the left and right habenula.

## Image analysis

Confocal imaging was carried out using a Leica TCS SP8 system with a 25x/0.95 NA PL FLUOROTAR water-dipping objective for live-imaging or 25x/0.95 NA PL IRAPO water-immersion objective with coverslip correction for fixed samples. Image analysis was performed using Fiji (ImageJ) and Imaris 7.7.1 (Bitplane) software. Images and figures were assembled using Adobe Photoshop and Adobe Illustrator.

## Acknowledgements

We thank all members of our labs in UCL and at the CBI Toulouse for support, the UCL and CBI Fish Facilities for help with fish maintenance and the zebrafish research community in London for fruitful discussions.

## Additional information

### Funding

| Funder | Grant reference number | Author |
|---|---|---|
| Wellcome | 104682/Z/14/Z | Stephen W Wilson |
| Wellcome | 099749/Z/12/Z | Ingrid Lekk |
| Fondation ARC pour la Recherche sur le Cancer | RA14P0040 | Patrick Blader |
| Fondation pour la Recherche Médicale | DEQ20130326466 | Patrick Blader |
| Agence Nationale de la Recherche | ANR-16-CE13-0013-01 | Patrick Blader |
| Fondation Fyssen | | Véronique Duboc |
| Fondation ARC pour la Recherche sur le Cancer | | Véronique Duboc |

The funders had no role in study design, data collection and interpretation, or the decision to submit the work for publication.

### Author contributions

Ingrid Lekk, Conceptualization, Data curation, Formal analysis, Supervision, Validation, Investigation, Visualization, Methodology, Writing—original draft, Project administration, Writing—review and editing, Created one of the sox1a mutant lines, Conducted phenotypic analyses in both sox1a mutant lines, Performed ablation and transplantation experiments; Véronique Duboc, Conceptualization, Formal analysis, Funding acquisition, Validation, Investigation, Methodology, Project administration, Created the main sox1a mutant line, Conducted initial phenotypic analysis; Ana Faro, Conceptualization, Supervision, Methodology, Project administration; Stephanos Nicolaou, Validation, Investigation, Visualization, Methodology; Patrick Blader, Conceptualization, Resources, Supervision, Funding acquisition, Investigation; Stephen W Wilson, Conceptualization,

Resources, Supervision, Funding acquisition, Investigation, Writing—original draft, Project administration, Writing—review and editing

## Author ORCIDs
Ingrid Lekk (iD) https://orcid.org/0000-0002-0286-8992
Patrick Blader (iD) http://orcid.org/0000-0003-3299-6108
Stephen W Wilson (iD) https://orcid.org/0000-0002-8557-5940

## Ethics

Animal experimentation: All experimental procedures at UCL were conducted under licence from the UK Home Office, following UK Home Office regulations and/or European Community Guidelines on Animal Care and Experimentation and were approved by animal care and use committees. Work at the CBI was performed in strict accordance with French and European guidelines. French veterinary service and national ethical committee approved the protocols in this study, with approval ID: A-31-555-01 and APAPHIS #3653-2016011512005922v6.

## Decision letter and Author response

Decision letter https://doi.org/10.7554/eLife.47376.017
Author response https://doi.org/10.7554/eLife.47376.018

## Additional files

### Supplementary files

• Transparent reporting form
DOI: https://doi.org/10.7554/eLife.47376.015

### Data availability

All data generated or analysed during this study are included in the manuscript and supporting files.

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
