## [Decision Letter]

Thank you for submitting your article "Sox1a mediates the ability of the parapineal to impart habenular left-right asymmetry" for consideration by *eLife*. Your article has been reviewed by three peer reviewers, and the evaluation has been overseen by a Reviewing Editor and Marianne Bronner as the Senior Editor. The following individual involved in review of your submission has agreed to reveal their identity: Cecilia B Moens (Reviewer #3).

The reviewers have discussed the reviews with one another and the Reviewing Editor has drafted this decision to help you prepare a revised submission.

Summary:

The paper of Lekk et al. studies the ability of a small group of cells of the embryonic zebrafish brain, called the parapineal, to impart left-right asymmetry in the habenula. For this, it uses a combination of genetic, imaging and embryological (ablation, transplantation) approaches, which are technically rigorous. These approaches are in some cases novel and elegant (e.g. parapineal transplants) while in others are used in a more systematic manner than previous studies giving attention to a relevant variable previously overlooked (e.g. role of "time" by temporally-directed cell ablations). These approaches (parapineal ablation and transplantation) together with the generation and analysis of the *sox1a^-/-^* mutant, provide new exiting data on the role of the parapineal in habenular asymmetry, also opening new questions in the field. Among the most novel results, the authors show that the parapineal (a) is not only required (as shown by previous studies) but also sufficient to induce the development of asymmetries in the habenula, at least, in the developmental windows used in the study; (b) regulates different steps of habenular asymmetry at different times (early role in neurogenesis and later role in axonal outgrowth and development of neurotransmitter domains); and (c) requires the function of *sox1a* to influence habenular asymmetry.

Essential revisions:

1) Neurogenesis in the *sox1a* mutant

The authors have shown by various criteria that the effect of ablating the parapineal is similar to the effect of *sox1a* loss. However, to make the argument (currently implied) that *sox1a* is required for accelerated habenular neurogenesis they should test the hypothesis directly.

2) Cell autonomy of *sox1a*

Transplants of *sox1a^-/-^* mutant parapineal cells into WT need to be performed, to provide a more direct assessment of the requirement of *sox1a* to induce habenular asymmetry by the parapineal, and also to test the cell autonomous vs. non-cell autonomous roles of *sox1a* in both habenular asymmetry and in the generation of axonal projections by the parapineal. In this same context, a better description of the morphology of parapineal axons could also help.

3) Cell-cell contact

This is not a required experiment, but if you have any data that would give insight it would be helpful. If not, then simply discussing this in the manuscript would be sufficient. Does the competence provided by *sox1a* requires cell-cell contact between parapineal and habenula? This can be achieved by generating mosaic parapineal (containing wt and *sox1a* cells) in a *sox1a^-/-^* context (by classical transplantation at blastula stages) and testing if the position of wt parapineal cells (in contact vs. not in contact with the Hb) influences the induction of *kctd12.1* expression.

---

## [Author Response]

Essential revisions:1) Neurogenesis in the sox1a mutantThe authors have shown by various criteria that the effect of ablating the parapineal is similar to the effect of sox1a loss. However, to make the argument (currently implied) that sox1a is required for accelerated habenular neurogenesis they should test the hypothesis directly.

We agree that an analysis of neurogenesis in the *sox1a* mutant would strengthen our study. Consequently, we carried out a BrdU birth-dating analysis in *sox1a* homozygous mutants compared to wild-type siblings and show a decrease in left habenula neurogenesis at 32 hpf upon loss of Sox1a function. This is similar to the change seen following parapineal ablations and confirms that signalling activity from the parapineal is lost in the absence of Sox1a function (Figure 6, first paragraph of the subsection “Asymmetric habenular neurogenesis is regulated by the parapineal” in the Results section).

2) Cell autonomy of sox1aTransplants of sox1a^-/-^ mutant parapineal cells into WT need to be performed, to provide a more direct assessment of the requirement of sox1a to induce habenular asymmetry by the parapineal, and also to test the cell autonomous vs. non-cell autonomous roles of sox1a in both habenular asymmetry and in the generation of axonal projections by the parapineal. In this same context, a better description of the morphology of parapineal axons could also help.

The importance of the parapineal in habenular asymmetry was first suggested in papers in 2003 yet this is the first study to have managed to show directly that parapineal cells can signal by transplanting them to an ectopic location. However, we would like to emphasise to the editor and reviewers just how technically challenging these few-cell transplants are. Hence, although we understand the reasoning to request transplants with mutant parapineal cells into wildtype embryos, we consider this a problematic experiment. Based on our extensive current results, the most likely outcome of this experiment would be that the mutant parapineal cells show no signalling activity and consequently no effect on habenular asymmetries. This would mean that we would have no clear read-out for the success of the experiment. Due to technical difficulties, multiple repetitions of the experiment in order to convincingly claim a lack of effect is not feasible.

Assessing the development of projections from *sox1a^-/-^*parapineal cells in a wild-type left-side environment would indeed be of interest. However, we are not able to perform this experiment within a reasonable time-frame as it would require us to generate lines of fish carrying multiple transgenes, most importantly a fluorescent reporter other than GFP to distinguish transplanted *sox1a^-/-^*parapineal cell projections from those of the endogenous parapineal at 4 dpf. Although we consequently cannot address the direct effects of Sox1a upon axon extension, this is not a key issue within the study.

Likewise, in experiments where wild-type parapineal cells were transplanted to the left side of *sox1a^-/-^*embryos, the rescue of the left habenula phenotype might also in turn have led to the rescue of the endogenous *sox1a^-/-^* parapineal axons at 4 dpf (see Figure 4A’). However, we cannot trace with confidence whether the extensive axonal arborisations stem solely from the transplanted (wild-type) or also from the endogenous (*sox1a^-/-^*) parapineal cells using the transgenic lines we have at our disposal.

We add extra text and an additional figure to discuss these points more thoroughly in the revised manuscript. The different morphology of the parapineal axons in *sox1a^-/-^*mutants compared to wild-type siblings is further discussed in the Results (subsection “The parapineal forms in *sox1a^-/-^* mutants”, first paragraph) and shown in an added figure (Figure 2—figure supplement 2).

3) Cell-cell contactThis is not a required experiment, but if you have any data that would give insight it would be helpful. If not, then simply discussing this in the manuscript would be sufficient. Does the competence provided by sox1a requires cell-cell contact between parapineal and habenula? This can be achieved by generating mosaic parapineal (containing wt and sox1a cells) in a sox1a^-/-^ context (by classical transplantation at blastula stages) and testing if the position of wt parapineal cells (in contact vs. not in contact with the Hb) influences the induction of kctd12.1 expression.

Unfortunately, it is not feasible to routinely label parapineal cells with blastula transplants due to its very small size. Furthermore, the parapineal has variable morphology during its migration and so it would be even more challenging, probably not feasible, to obtain sufficient numbers of mosaic parapineals to draw any convincing conclusions. We also note that, to date, there has not been any study that has comprehensively analysed the extent of contacts between parapineal and habenular cells during normal development.

Further discussion regarding possible cellular mechanisms of parapineal-habenula interaction are included in the Results section, subsection “Wild-type parapineal cells induce left habenula characteristics in *sox1a^-/-^* mutants”, second paragraph (related to Figure 4) and at the end of the first section of the Discussion section.